# Bovine κ-Casein Fragment Induces Hypo-Responsive M2-Like Macrophage Phenotype

**DOI:** 10.3390/nu11071688

**Published:** 2019-07-23

**Authors:** Richard Lalor, Sandra O’Neill

**Affiliations:** Fundamental and Translational Immunology Group, Dublin City University, Dublin 9, Ireland

**Keywords:** kappa-casein, sodium caseinate, macrophage, immunosuppression, nutraceutical

## Abstract

Immunomodulatory nutraceuticals have garnered special attention due to their therapeutic potential for the amelioration of many chronic inflammatory conditions. Macrophages are key players in the induction, propagation and resolution of inflammation, actively contributing to the pathogenesis and resolution of inflammatory disorders. As such, this study aimed to investigate the possible therapeutic effects bovine casein derived nutraceuticals exert on macrophage immunological function. Initial studies demonstrated that sodium caseinate induced a M2-like macrophage phenotype that was attributed to the kappa-casein subunit. Kappa-casein primed macrophages acquired a M2-like phenotype that expressed CD206, CD54, OX40L, CD40 on the cell surface and gene expression of Arg-1, RELM-α and YM1, archetypical M2 markers. Macrophages stimulated with kappa-casein secreted significantly reduced TNF-α and IL-10 in response to TLR stimulation through a mechanism that targeted the nuclear factor-κB signal transduction pathway. Macrophage proteolytic processing of kappa-casein was required to elicit these suppressive effects, indicating that a fragment other than C-terminal fragment, glycomacropeptide, induced these modulatory effects. Kappa-casein treated macrophages also impaired T-cell responses. Given the powerful immuno-modulatory effects exhibited by kappa-casein and our understanding of immunopathology associated with inflammatory diseases, this fragment has the potential as an oral nutraceutical and therefore warrants further investigation.

## 1. Introduction

Nutraceutical is a term derived from “nutrition” and “pharmaceutical” that is applied to products that are isolated from herbal, dietary supplements and functional foods such as dairy, cereals and beverages which, beyond nutritional value, possess physiological benefits to improve health or prevent chronic diseases [1,2]. Milk, in particular, has a great potential to be used commercially as a source of nutraceuticals, as the production and consumption of milk products has increased [3] and the proteins and peptides derived from milk have already been shown to display an array of bioactive properties, including anti-tumour, anti-microbial, anti-oxidant, opioid, ACE-inhibitory and immune-modulatory activity [4,5,6].

Mounting evidence suggests that the number of individuals who suffer from chronic inflammatory conditions has increased, with an estimated prevalence of 5 to 7% among developed countries [7]. Research into the use of immuno-modulatory bioactive protein-based nutraceuticals has gained interest due to their potential use as a dietary intervention strategy in the treatment of many immune-related diseases. Currently, the most frequently used strategies involves the use of drugs that slow the progression of specific diseases, however, they can often have unforeseen and potentially harmful side effects which can outweigh their benefits [8]. In this context, immune modulation via dietary supplementation with bioactive nutraceuticals may represent a viable alternative as they display beneficial properties through the stimulation or inhibition of certain immune functions. These compounds also exhibit low toxicity, are easily degraded and tend not to accumulate in bodily tissues [9,10]. Moreover, bioactive-based proteins and peptides are generally small enough to allow efficient delivery/adsorption and ensure a low likelihood of triggering undesirable immune responses [10,11,12,13].

A great body of evidence indicates that bovine milk-derived proteins have the potential to modulate immune function in a number of species [14]. One of the most abundant milk proteins, casein (CAS), its individual subunits (αs_1_, αs_1_, β and κ) and hydrolysed derivatives exert immuno-suppressive properties and ameliorate inflammatory diseases in mice and humans [15,16,17]. Previous studies have particularly focused on the immuno-modulatory effect that these bioactive proteins and peptides exhibit on antigen presenting cells (APCs), which are heavily implicated in the development of these chronic inflammatory diseases [18,19,20]. Macrophages represent a population of APCs, distributed throughout bodily tissues, which can exhibit effecter functions that enable them to activate or dampen immune responses by the release of immuno-stimulatory factors, such as cytokines, and can present antigen in-situ, driving adaptive immunity [21]. CAS and its hydrolysed derivatives reduced macrophage phagocytic function and suppressed the production of reactive oxygen and nitrogen species in response to inflammatory stimuli [22,23]. These derivatives also attenuate NF-κB activation via upregulation of heme oxygenase-1 [24] inducing the differentiation of hematopoietic precursors into macrophages that fail to produce TNF-a on LPS stimulation [25,26]. Thus, a CAS-derived bioactive nutraceutical with immuno-modulatory properties which affects macrophages would be of great interest due to their prominent role in both innate and adaptive immunity.

While extensive research on CAS, its subunits and their derivatives has clearly demonstrated their potential use as immunomodulatory compounds [22,23,24,25,26,27], there is a dearth of research on the mechanism by which these compounds dampen inflammatory responses. Therefore, more studies are required to define the cellular phenotype which not only affects their immediate effector functions but can also heavily influence their ability to initiate and propagate adaptive immune responses [28]. In particular, studies should focus on understanding the mechanisms by which these compounds exert their suppressive effects. This study will therefore further examine the use of immuno-modulatory milk casein based bioactive proteins in order to advance our understanding of the impact they have on key innate immune cells, potentially leading to the discovery of a new viable alternative to the use of pharmaceuticals in influencing health for the prevention and treatment of chronic diseases.

## 2. Experimental Section

### 2.1. Reagents and Materials

Sodium caseinate (CAS, the CAS content of protein is a minimum of 95%) was provided by Teagasc (Moorepark, Ireland). Alpha (α-CAS), beta (β-CAS) and kappa (κ-CAS) caseins were purchased from Sigma-Aldrich (St. Louis, MO, USA). Lipopolysaccharide (LPS) from *E. coli*, serotype R515 was purchased from Enzo Life Sciences (Exeter, UK). All the antibodies used in this investigation were obtained from commercial sources. Mouse monoclonal inhibitor of κB-α (IκBα, No. ab32518) antibody was purchased from Abcam (Cambridge, UK). Mouse monoclonal β-actin (No. 643802) antibody was purchased from Biolegend (San Diego, CA, USA). Horseradish peroxidase-conjugated anti-species (mouse and rabbit) secondary antibodies were purchased from Bio-Rad Laboratories (Hercules, CA, USA). Macrophage-colony stimulating factor (M-CSF) was obtained from a M-CSF producing cell line L929 (LGC Standards, Middlesex, UK), based on a previously established method [29]. Cell culture material was purchased from Biosciences (Dun Laoghaire, Ireland).

### 2.2. Animals and Ethics

C57BL/6J mice aged 6–8 weeks were purchased from Charles River UK Ltd. (Kent, UK) and kept under specific pathogen-free conditions at the DCU Bioresources unit. All the mice were housed according to the Health Products Regulatory Authority guidelines and the standard operating procedure approved by the institutional Animal Welfare Body were strictly adhered too. Ethical permission for the use of animals was approved by the Department of Health or Health Products Regulatory Authority and Dublin City University ethics committee (licence numbers B100/2833, DCUREC/2010/033). All the procedures involving animals were only performed by licensed personnel.

### 2.3. Generation of Bone Marrow-Derived Macrophages

Bone marrow-derived macrophages (BMMΦ) were differentiated using a previously described method [29]. C57BL/6JCrl mice purchased from Charles River (Kent, UK) were sacrificed by cervical dislocation. Bone marrow from femurs and tibias were extracted and seeded in petri dishes at a cell density of 10 × 10^6^ cells/10 mL in RPMI supplemented with 20 ng/mL M-CSF, 10% fetal calf serum, 1% l-glutamine, and 100 μ/mL penicillin/streptomycin. The media was replenished on day 3. On day 6, non-adherent cells were removed by washing with PBS. Adherent cells were detached with an accutase detachment solution (Biosciences, Dun Laoghaire, Ireland). Macrophage purity was analysed by flow cytometry, with >95% of the population identified as macrophages on the basis of double positive expression of both CD11b (Biolegend, No. 101215) and F4/80 (Biolegend, No. 123115).

### 2.4. Generation of Monocyte Derived Human Macrophages

Monocyte derived human macrophages (hMφ) were differentiated using a previously described method [30]. Peripheral blood mononuclear cells (PBMCs) were isolated from the buffy coats of healthy donors obtained from the Irish blood transfusion service (St James’s hospital, Dublin) by density gradient centrifugation using Histopaque-1083 (Sigma-Aldrich, St. Louis, MO, USA). CD14^+^ monocytes were isolated from PBMCs using a magnetic activated cell sorting positive selection CD14^+^ isolation kit (Miltenti, Bergisch Gladbach, Germany). The monocytes were seeded at a cell density of 1 × 10^6^ cells/mL in RPMI media supplemented with 10% (*v*/*v*) human AB serum (Invitrogen, Carlsbad, CA, USA), 1% l-glutamine, and 100 μ/mL penicillin/streptomycin. The 20 × 10^6^ cells were transferred to a 75 cm^2^ vented adherent flask (Sarstedt, Nümbrecht, Germany) and cultured at 37 °C and 5% CO_2_. The culture media was renewed every 3 days and monitored morphologically for differentiation. Non-adherent cells were removed on day 7 by aspirating all media. On day 14, cells adherent hMφ were harvested using 0.25% trypsin-EDTA solution (Sigma-Aldrich, St. Louis, MO, USA) at room temperature.

### 2.5. Cell Activation

Cells were pre-treated with the indicated concentrations of CAS 2.5 h prior, at the same time or 2.5 h post toll like receptor (TLR) agonist (Alexis Biochemicals, San Diego, CA, USA) stimulation. The TLR agonists used were TLR4 agonist, lipopolysaccharide (LPS) (100 ng/mL), TLR2 agonist, peptidoglycan (PGN) (5 μg/mL), TLR7 agonist, loxoribine (LOX) (0.5 mM), or TLR9 agonist, synthetic oligonucleotides containing CpG motifs (CpG) (2 μM). TLR agonist or PBS stimulations alone were used as positive and negative controls respectively.

To elucidate if heme oxygenase-1 was involved in the signalling pathways by which casein exerts its effects, cells were cultured with chemical antagonists of heme oxygenase-1; Zinc-protoporphyin-9 (ZnPPIX) (Santa Cruz Biotechnology, Santa Cruz, CA, USA) at the indicated concentrations, 30 min prior to addition of caseins.

To elucidate if the whole protein or an active fragment released after cellular processing was involved in the effects exerted by κ-CAS, cells were cultured with a Halt™ protease inhibitor cocktail (1:500 (*v*/*v*)) (Thermo Scientific, Waltham, MA, USA), 30 min prior to addition of caseins.

### 2.6. CD4^+^ T-Cells Co-Culture

Spleens from C57BL/6JCrl (Charles River, Kent, UK) mice were extracted and spleenocytes obtained by passage of the spleen through a 40 μm filter (Sarstedt, Nümbrecht, Germany) using the plunger from a sterile 1 mL syringe (Sarstedt, Nümbrecht, Germany). CD4^+^ T-cells were isolated from spleenocytes using a negative selection CD4^+^ isolation kit (Stemcell, Vancouver, BC, Canada) and were only used if the purity was determined to be >95% CD4^+^ (Biolegend, No. 116005) by flow cytometry. Pre-stimulated cells were washed and co-cultured with CD4^+^ T-cells at a ratio of 1:4 in RPMI media supplemented with 10% fetal calf serum, 1% l-glutamine, and 100 μ/mL penicillin/streptomycin on cell culture plates pre-coated overnight with anti-CD3 (1 µg/mL) (R & D systems, Minneapolis, MN, USA).

For human allogenic T-cell co-cultures, the T-cells were isolated from buffy coats by a magnetic activated cell sorting positive selection CD4^+^ isolation kit (Miltenti, Bergisch Gladbach, Germany). Pre-stimulated cells were washed and co-cultured with human CD4^+^ T-cells at a ratio of 1:10 in RPMI media supplemented with 10% (*v*/*v*) human AB serum, 1% l-glutamine, and 100 μ/mL penicillin/streptomycin on cell culture plates pre-coated overnight with human anti-CD3 (1 µg/mL) (R & D systems, Minneapolis, MN, USA).

### 2.7. Flow Cytometry

Cells were harvested, re-suspended in ice cold flow cytometry buffer (PBS supplemented with 2% fetal calf serum and 1 mM EDTA) and incubated with the fluorochrome labelled anti mouse CD40 (Biolegend, No. 124609), CD54 (Ebioscience, No. 12-0541), CD206 (Biolegend, No. C068C2), OX40L (Biosciences, No. 12-5905) antibodies for 30 min at 4 °C in the dark. After incubation, the cells were washed with flow cytometry buffer to remove any unbound fluorochrome labelled antibodies and processed on the flow cytometer FACs Aria (Becton Dickinson, Franklin Lakes, NJ, USA). Data were analysed using FlowJo software (Treestar, Woodburn, OR, USA). Unlabeled, single fluorocrhome labelled and fluorocrhome labelled isotype antibodies were used as controls for non-specific staining of cells and compensation.

### 2.8. Polymerase Chain Reaction

Total RNA were extracted from cultured cells using a RNA isolation kit (Roche Diagnostics, West Sussex, UK) according to the manufacturer’s guidelines. cDNA was synthesized from the isolated RNA using a transcriptor first strand cDNA synthesis kit (Roche Diagnostics, West Sussex, UK) according to the manufacturer’s guidelines. Synthesized cDNA was used as a template for polymerase chain reaction (PCR) using primers (all from Invitrogen, Carlsbad, CA, USA) specific for *Arg 1*, *Ym-1*, *iNOS*, *RELM α*, and *β-actin* (Table 1). Samples were maintained at 95 °C for 1 min as an initial step, followed by 60 °C for 30 s, and finally 72 °C for 1 min. These amplification cycles were carried out 40 times. The process was preceded by a denaturation phase at 95 °C for 5 min and a final extension phase of 72 °C for 5 min. PCR products were electrophoresed on 1% agarose gels with SYBRSafe (Invitrogen, Carlsbad, CA, USA) as gel stain.

### 2.9. Sodium Dodecyl Sulphate Polyacrylamide Gel Electrophoresis

Sodium dodecyl sulphate polyacrylamide gel electrophoresis (SDS-PAGE) was performed based on a previously established method [31]. Cells were harvested, washed with PBS and lysed with RIPA buffer (Sigma-Aldrich, St. Louis, MO, USA) containing 1X Halt protease and phosphatase inhibitor cocktail (Thermo Scientific, Waltham, MA, USA). After centrifugation at 6000× *g* for 10 min at 4 °C, the protein was quantified using the bicinchoninic acid assay (Thermo Scientific, Waltham, MA, USA). Protein from each sample (20 μg) was loaded, subjected to SDS-PAGE and transferred onto PVDF membranes (Millipore, Billerica, MA, USA). Membranes were blocked with PBS-T solution (PBS supplemented with 0.05% Tween-20) supplemented with 5% skimmed milk (Marvel) at room temperature for 2 h and subsequently incubated overnight at 4 °C with the primary antibodies anti-IκBα (Abcam, No. ab32518) or β-actin (Biolegend, No. 643802). After washing with PBS-T solution, the PVDF membranes were incubated with horseradish peroxidase conjugated secondary antibody (Bio-rad, Hercules, CA, USA) for 1 h. Immunolabeled proteins were washed with PBS-T solution and visualized with a chemiluminescent HRP substrate (Millipore, Burlington, MA, USA), on a G-Box imaging system (Syngene, Cambridge, UK). Protein bands were quantified using ImageJ analysis software (SciJava consortium). The levels of protein were normalised to the control gene β-actin.

### 2.10. Cytotoxicity Assays

Resazurin assays were used to determine the cytotoxicity of compounds. Briefly, cells were treated with respective stimulations and at the end time point, incubated with 0.15 mg/mL resazurin salt (Sigma-Aldrich, St. Louis, MO, USA) for an additional 6 h. The absorbance values were recorded using a TECAN genios microplate reader (Tecan Genios). The cytotoxic effects were measured and compared to vehicle stimulated controls. The cytotoxic effects of stimulants on the cells were also measured using the Annexin V-FITC apoptosis detection kit I (BD Biosciences, San Jose, CA, USA) and analysed by flow cytometry.

### 2.11. Statistical Analysis

All data was analysed for normality prior to statistical testing by Prism^®^ 6.1 software (GraphPad software Inc.). Where multiple group comparisons were made, data was analysed using one- or two-way ANOVA. For comparisons between two groups, the Student’s *t* test was used. In all the tests, *p* < 0.05 were deemed significant.

## 3. Results

### 3.1. NaCAS Induces an M2-Like Macrophage Phenotype that Exhibits a Reduced Responsiveness to LPS Stimulation

Previous studies [26] have examined the effect sodium caseinate (a soluble form of CAS, NaCAS) exerted on TLR4 induced cytokine responses in bone marrow derived macrophages (BMMφ). Similar to these studies we also observed a dose-dependent reduction in LPS-induced TNF-α (Figure 1a **, *p* ≤ 0.01 *, *p* ≤ 0.05) and IL-10 (Figure 1b *, *p* ≤ 0.05) production by NaCAS treatment. We further examined the effect that NaCAS exerts on macrophages by investigating any influence CAS has on macrophage phenotypes, which can acquire either a classically activated M1 or an alternatively activated M2 phenotypic state [21].

We sought to compare the differences in phenotypic markers between BMMφs stimulated with NaCAS to cells differentiated into an M1 (stimulated with IFN-γ or IFNγ & LPS), M2a (stimulated with IL-4) and M2c (stimulated with PGE_2_) phenotypes. Our results demonstrated that NaCAS induced the expression of M2-associated genes *Arg-1*, *Ym-1* and *RELM-α*, similar to an M2 phenotype induced by IL-4 & PGE_2_ controls (Figure 2a). No induction of *iNOS* was observed, a marker of M1 activation, except for the relevant IFN-γ and LPS stimulated control, suggesting that NaCAS selectively induces an M2-like macrophage phenotype. Further characterization of the phenotype induced by NaCAS was examined by comparing the cell surface marker expression of NaCAS treated BMMφs to differentiated M1, M2a and M2c macrophage phenotypes. BMMφ stimulated with NaCAS resulted in a significant increase in M2a associated CD206 (Figure 2b **, *p* ≤ 0.01) and MGL (Figure 2c *, *p* ≤ 0.05). However, NaCAS did not induce any increased expression of Dectin-1 (Figure 2d), another extracellular receptor often associated with M2a macrophages. Interestingly, NaCAS also significantly increased CD54 (Figure 2e **, *p* ≤ 0.01), a marker normally associated with classical M1 activation. In summary, NaCAS induced a M2-like phenotype that expressed a mixed M1 and M2 extracellular marker repertoire and exhibited a reduced responsiveness to TLR4 stimulation.

### 3.2. κ-CAS Is the Subunit Responsible for the Suppression of LPS-Induced Cytokine Responses and the Induction of M2 Related Genes in BMMφ

CAS is comprised of four protein subunits: α_s1_-, α_s2_-, β- and κ-CAS [32] and therefore, we next sought to determine if a single subunit or multiple subunits are involved in the induction of this M2-like phenotype. κ-CAS was shown to significantly attenuate LPS-induced TNF-α (Figure 3a **, *p* ≤ 0.01) and IL-10 (Figure 3b **, *p* ≤ 0.01). However, while α- and β-CAS also secreted significantly less IL-10 in response to LPS, no reduction in TNF-α secretion was observed. Furthermore, α- and β-CAS treatment alone in the absence of LPS-induced the secretion of TNF-α from BMMφs. Similar to the results we had previous obtained for NaCAS, κ-CAS abrogated the release of TNF-α (Figure 3c **, *p* ≤ 0.01 *, *p* ≤ 0.05) and IL-10 (Figure 3d **, *p* ≤ 0.01 *, *p* ≤ 0.05) in a dose-dependent manner.

When examining influences on macrophage phenotypes, κ-CAS was also shown to induce the M2 associated genes *Arg-1*, *RELM-α* and *Ym-1* (Figure 4a), while no induction of the genes associated with M1 or M2 phenotypes was observed for α- or β-CAS treatment (data not shown). Further characterization revealed that κ-CAS also increased the expression of the extracellular receptors CD206 (Figure 4b **, *p* ≤ 0.01) and CD54 (Figure 4c **, *p* ≤ 0.01), similar to the results we had previous obtained for NaCAS, however no increases in MGL were observed (data not shown). Interestingly, significant differences in the expression of the co-stimulatory markers, OX40L (Figure 4d *, *p* ≤ 0.05) and CD40 (Figure 4e *, *p* ≤ 0.05), were detected by κ-CAS treatment.

### 3.3. κ-CAS Abrogates NF-κB Activation

Having observed the effects κ-CAS exerted on cytokine secretions in response to LPS stimulation, we sought to investigate if the inhibition of NF-κB signaling was involved in this phenomenon. We examined the effects κ-CAS exhibited on the degradation of inhibitory IkB proteins, a key process in the NF-κB signaling cascade required for the induction of pro-inflammatory genes and the production of cytokines in response to inflammatory stimuli like LPS [33]. PBS and κ-CAS alone showed no significant differences in protein band intensity, indicating that no degradation of the IkB-α protein had occurred. The time point of optimal IκB-degradation was deduced to be at 15 min after LPS treatment, which led to a significant reduction in the levels of IkB-α protein. We demonstrated that pre-treatment with κ-CAS prior to LPS stimulation led to a reduction in the degradation of the IkB-α protein compared to the LPS treated control (Figure 5a). Densitometric analysis of the blots also revealed that there was a significant difference between the samples pre-treated with κ-CAS and subsequently stimulated with LPS and LPS only treated controls (Figure 5b *, *p* ≤ 0.05), which would infer that κ-CAS may partially abrogate NF-κB signaling.

### 3.4. An Active Fragment of κ-CAS Is Responsible for Its Effect Which Targets Multiple TLRs Signalling Pathways

Several peptides derived from the proteolysis of κ-CAS exert immuno-suppressive activity, notably the C-terminal fragment of κ-CAS GMP and its hydrolysate derivatives [22,23,24,27,34,35]. GMP- and GMP-derived hydrolytates were also shown to inhibit LPS-mediated inflammatory responses in the macrophages by attenuating NF-κB activation [35], attributed to the upregulation of heme oxygenase-1 [24]. Given that we attained similar results with κ-CAS, we next examined if intact κ-CAS or a fragment accounted for the observed activity and if any effects were likely due to GMP and the upregulation of heme oxygenase-1. We demonstrated that a protease inhibitor cocktail containing 4-(2-aminoethyl)-benzenesulphonyl fluoride, aprotinin, leupeptin, bestatin, pepstatin A and E-64 which blocked the possible hydrolysis of κ-CAS by cell proteases, reversed the suppressive effects κ-CAS exhibited on LPS-induced TNF-α (Figure 6a) production. Moreover, the inhibition of heme oxygenase-1 by zinc Protoporphyrin-9 (ZnPPIX) did not restore inflammatory cytokine release (Figure 6b). These results suggest that the suppressive mechanism exerted by κ-CAS on LPS-induced cytokine production is due to a fraction of κ-CAS with immune-modulatory activity that exerts its effects independent of heme oxygenase-1.

We also sought to investigate if the time of exposure to κ-CAS had any impact on its effect on cytokine production in LPS stimulated BMMφ and if κ-CAS targeted other TLRs. κ-CAS significantly suppressed the production of TNF-α (Figure 6c **, *p* ≤ 0.01) in LPS stimulated BMMφs when added prior to (−2.5 h) LPS stimulation as shown previously. However, BMMφ were equally as suppressed when treated with κ-CAS simultaneously (0 h) or after (+2.5 h) LPS stimulation. No significant differences in cytokine reduction were detected between the exposure times. κ-CAS was also observed to significantly suppress the secretion of TNF-α and IL-10 (Figure 6d **, *p* ≤ 0.01) in response to TLR2 (PGN), TLR7 (LOX) and TLR9 (CpG) agonists. Therefore, κ-CAS targets multiple TLR signaling pathways and can exert its effects prior and post exposure to inflammatory stimuli.

### 3.5. κ-CAS Sequesters the T-cell Priming Capacity of Macrophages

While previous studies have examined the effects κ-CAS and its hydrolysate derivatives have on T-cell activity directly, the ability of κ-CAS stimulated macrophages to modulate T-cells has not been previously shown. Macrophages can present antigen to responsive T-cells, participating directly in the generation of adaptive immune responses [36]. Considering κ-CAS significantly upregulated costimulatory marker expression in BMMφ (Figure 4), we investigated the impact these cells had on wider inflammatory process by examining their interaction and priming of T-cells. BMMφs stimulated with κ-CAS induced significantly less IFN-γ (Figure 7a *, *p* ≤ 0.05) and IL-2 (Figure 7b **, *p* ≤ 0.01) compared to control BMMφ stimulated with PBS. However, no significant differences in the levels of IL-13 or IL-10 were observed (data not shown). Moreover, CD4^+^ T-cells co-cultured with κ-CAS treated BMMφ showed no increases in the expression of the cell surface markers CTLA-4 (Figure 7c) or PD-1 (Figure 7d) compared to anti-CD3 anergic controls, which would suggest that κ-CAS-treated macrophages have a significantly reduced capacity to elicit T-cell responses, independent of inducing anergic T-cells.

### 3.6. The Immunosuppressive Effects Exerted by κ-CAS Are Transferable in Human Cells

Having observed κ-CAS capacity to modulate murine macrophage phenotypes, cytokine responses and T-cell priming capabilities, we next examined if these effects were transferable to humans using human monocyte derived macrophages (hMφ). Similar to the results obtained from murine BMMφ, κ-CAS significantly attenuated LPS-induced TNF-α secretion in hMφ (Figure 8a **, *p* > 0.01). Moreover, similar to the effects observed in mice, κ-CAS was also shown to reduce the capacity of monocytes (macrophage precursors) to induce IL-2 from allogenic CD4^+^ T-cells co-cultures (Figure 8b *, *p* ≤ 0.05). However, no significant differences in the levels of IFN-γ, IL-13 or IL-10 were observed (data not shown). As a reduction in IL-2 is often associated with anergy or apoptosis, the expression of the extracellular anergic marker CTLA4 and the uptake of propidium iodide as a measure of cell viability was assessed. Co-cultured CD4^+^ T-cells exhibited no upregulation of CTLA4 (Figure 8c) or uptake of propidium iodide (Figure 8d). This data would infer that κ-CAS acts similarly on human macrophages, attenuating their responsiveness to inflammatory stimuli and significantly reducing their capacity to elicit T-cell responses, independent of anergic mechanisms.

## 4. Discussion

This study sheds new light on the immunomodulatory effects exhibited by NaCAS and κ-CAS on macrophages, key cells involved in the initiation and control of inflammation [37,38] of murine and human origin. Herein, we presented novel evidence which suggests that intact NaCAS induced an M2-like phenotype in macrophages, as it was shown to induce the M2 related genes *Arg-1*, *RELMα* and *YM-1* [21]. We also observed an upregulation of extracellular receptors, CD206 and MGL, other hallmarks of M2a macrophage phenotypes [39]. Interestingly, dectin-1, another receptor generally associated with M2a primed macrophages, was not upregulated. NaCAS increased the surface expression of the CD54, which has been shown to be important in cell to cell communication, signalling and mediating cell-cell or cell-extracellular matrix attachment [40,41]. While the induction of CD54 in macrophages has been generally associated with pro-inflammatory M1-like activation [42], its overexpression has been shown to promote M2 polarization [43]. Moreover, it has been suggested that CD54 is more a surrogate marker of APC activation rather than an indicator of its inflammatory status [42,44]. This would suggest that NaCAS induces a M2-like phenotype which does not strictly adhere to the described M1/M2 paradigm commonly cited in the literature [21]. Many studies have shown that M2-like macrophages phenotypes can be induced by non-classical stimuli. Endothelin-1, a pro-fibrotic peptide molecule released by endothelial cells, was shown to induce M2-like phenotypic characteristics [45] while helminth derived tegmental proteins and excretory/secretory products were also shown to induce M2-like phenotypes in macrophages [46,47]. Given that macrophages exhibit a plasticity of function which can be polarised based on exposure to external stimuli, a bioactive nutraceutical with immuno-modulatory properties, which affects a macrophages phenotype, and subsequent functionality would be of interest due their prominent role in both inflammatory processes and immune suppression.

Similarly to previous studies, we also observed the suppressive effects intact NaCAS exerted on LPS-induced cytokine secretion from macrophages [25,26,48,49]. We examined if a single subunit or multiple subunits were involved in the induction of this M2-like phenotype and the suppression of cytokine responses. Our study demonstrated that α- and β-CAS did not replicate the effects of NaCAS but instead enhanced the expression of TNF-α, a cytokine implicated in immune pathogenesis of many inflammatory disorders [50]. The induction of inflammatory activity by β-CAS conforms to an observation in the literature which demonstrated that β-CAS enhanced the production of oxidant species [22] and significantly induced the production of pro-inflammatory cytokine from macrophages [51]. However, in contrast to our observation, previous reports suggest that αs_1_-CAS reduced macrophage phagocytic function and suppressed the production of reactive oxygen and nitrogen species in response to inflammatory stimuli in a dose-dependent manner [22,23]. Moreover, αs_1_-CAS-derived peptides were shown to inhibit matrix metalloproteinase 9 activity, an enzyme involved in the induction of inflammatory cytokines [52,53]. The commercially available α-CAS used in this study is composed of a mixture of the αs_1-_ and αs_2-_CAS subunits which may account for the differences observed in this study. Unlike α- and β-CAS, the κ-CAS subunit was demonstrated to prime a similar M2-like phenotypic profile to that observed by the intact whole protein and also attenuated the ability of macrophages to produce of TNF-a and IL-10 in response to LPS. This would infer that most of the suppressive effects exhibited by the intact protein were due to the κ-CAS subunit, which is able to override any inflammatory effects exerted by α- and β-CAS, which do not display modulatory properties that would potentially be of benefit in the amelioration of inflammatory conditions.

Our observations are supported by other studies which reported on the immuno-suppressive properties exhibited by κ-CAS, such as reducing phagocytic function and suppressing the production of reactive oxygen and nitrogen species in response to inflammatory stimuli in murine macrophages [22,23]. We determined that proteolytic cleavage by cell proteases was required for the release of an active component responsible for the observed effects and not the whole κ-CAS subunit. Multiple other studies attributed the inhibitory activity of κ-CAS to the C-terminal fragment GMP and its derivatives [24,35]. One of the mechanisms proposed by which this occurred was found to be due to the upregulation of heme oxygenase-1, which, when inhibited, resulted in the restoration of inflammatory cytokine expression and NFκB activity [24]. While we attained similar results with κ-CAS, which was also shown to abrogate LPS-mediated inflammatory cytokine release and NFκB activation, the inhibition of heme oxygenase-1 did not restore inflammatory cytokine release. Our data also indicated that κ-CAS exerted its suppressive effects independent of when the TLR ligand was added and affected multiple other TLR pathways, suggesting that κ-CAS does not compete with LPS for binding and that κ-CAS does not exert its effects through the TLR, another mechanism by which GMP and its derivatives attenuated inflammatory cytokine release [35]. Given these differences, we can deduce that the results we obtained for κ-CAS are unlikely to be due to the GMP fragment but another novel fraction of κ-CAS with immune-modulatory activity that requires cleavage from the subunit prior to activation.

The state of activation and maturation of APCs like macrophages determines their ability to interact with T-cells, influencing the type of immune response initiated [54]. We demonstrated that κ-CAS primed macrophages exhibited a significantly reduced capacity to activate CD4^+^ T-cells and their subsequent release of IFN-γ, a cytokine strongly associated with T_H_1 adaptive immune responses [55]. Moreover, we also observed a significant reduction in the production of IL-2. Generally, a reduced signalling strength via the downregulation of co-stimulatory receptor interactions can result in the suppression of T-cell responses, however κ-CAS was shown to upregulate the co-stimulatory receptors CD40 and OX40L, which are generally associated with the induction of T-cell responses [56,57,58].

Other possible receptors, such as CD54 or CD206, could be responsible for the suppression of T-cell responses by κ-CAS activated macrophages. We demonstrated that CD54, an adhesion receptor, was upregulated by κ-CAS in macrophages. CD54 is known to be involved in APC-T-cell communication [59]. However, more recently CD54 expression on macrophages was shown to have immunosuppressive function at inflammatory sites, dampening the immune response [40]. This would suggest that CD54 expression on macrophages can exhibit stimulatory and regulatory properties. κ-CAS treatment also upregulated the CLR CD206. Aldridge & O’Neill demonstrated that the upregulation of CD206 on APCs can exert suppressive effects on T-cell cytokine responses in in-vitro co-culture [60]. The lack of inflammatory immuno-stimulatory factors produced by κ-CAS macrophages may account for the reduced T-cell responses. APCs expressing some costimulatory molecules but only low levels of inflammatory cytokines, such as TNF-α and IL-12, have also been reported to exert regulatory effects on T-cell responses [61].

Alternatively, activation through anergic pathways results in poor production of IL-2, the loss of proliferation and renders T-cells hypo-responsive [62]. However, CD4^+^ T-cells cultured with κ-CAS treated macrophages did not display either of the prominent extracellular surface markers associated with anergy CTLA4 and PD-1 [63]. Previous studies observed that intact κ-CAS and GMP significantly inhibited the mitogen-induced proliferative response of mouse spleen lymphocytes and Peyer’s patch cells [64,65]. However, our results also suggest that exposure to κ-CAS interferes with the ability of macrophages to induce T-cell responses given the lack of IL-2 required for the induction of a robust adaptive immune response [66].

The immunomodulatory effects exhibited by whole CAS protein, its subunits and hydrolysated derivatives have only been documented in a limited amount in previous studies using human cells. Similarly to the results obtained using murine cells, κ-CAS effects were shown to be transferable in human cells, attenuating hMφ responsiveness to inflammatory stimuli. Previous studies have only reported on the effects GMP exhibited on human macrophage-like cell lines, enhancing proliferation and phagocytic activity [67] and attenuated LPS responses in human colorectal tumor cells [68]. κ-CAS treated monocytes (macrophage precursors), when co-cultured with T-cells, were also shown to exhibit a significantly reduced capacity to activate CD4^+^ T-cells and their subsequent release of IL-2, independent of markers classically associated with anergy. This would indicate that κ-CAS acts similarly on human cells, attenuating their responsiveness to inflammatory stimuli and reducing their capacity to elicit T-cell responses by a similar mechanism that suppresses their ability to induce IL-2 production.

## 5. Conclusions

In conclusion, the results obtained in the present study demonstrate that κ-CAS induces a M2-like phenotype in macrophages, which are hypo-responsive to TLR induced cytokine production, via the abrogation of the NFκB pathway. These cells prime T-cell to induce significantly less T_H_1 associated pro-inflammatory cytokines and IL-2, sequestering the ability of κ-CAS treated macrophages to elicit adaptive immune responses in a non-anergic mechanism. These effects were transferable using human cells that were also rendered hypo-responsive to TLR stimulation and exhibited a reduced capacity to induce T-cell responses. Given the powerful immune-modulatory effects exhibited by κ-CAS on macrophages, key cells involved in the initiation and control of inflammation, further study is warranted to elucidate the sequence of the fragment responsible for these observed effects and lead to its development and use as a novel immune therapeutic in the treatment of inflammatory diseases.

## Figures and Tables

**Figure 1 nutrients-11-01688-f001:**
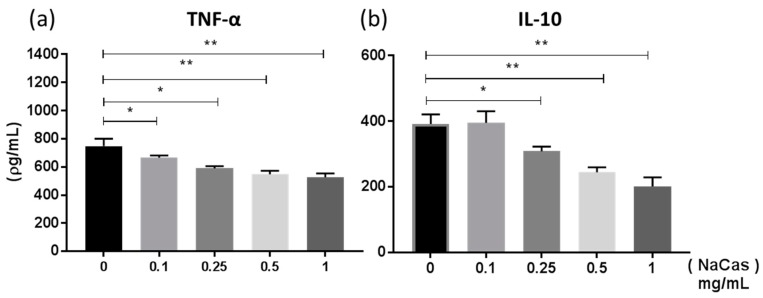
BMMφ were pre-treated with NaCAS (0.1–1 mg/mL) and subsequently stimulated in the presence or absence of LPS (100 ng/mL) for 18 h. Supernatants were analysed for the secretion of TNF-α (**a**) and IL-10 (**b**) by ELISA. Results are expressed as mean ± SD of at least three independent experiments. *p*-values were calculated using one-way ANOVA. *, *p* ≤ 0.05, **, *p* ≤ 0.01 compared to the PBS control group.

**Figure 2 nutrients-11-01688-f002:**
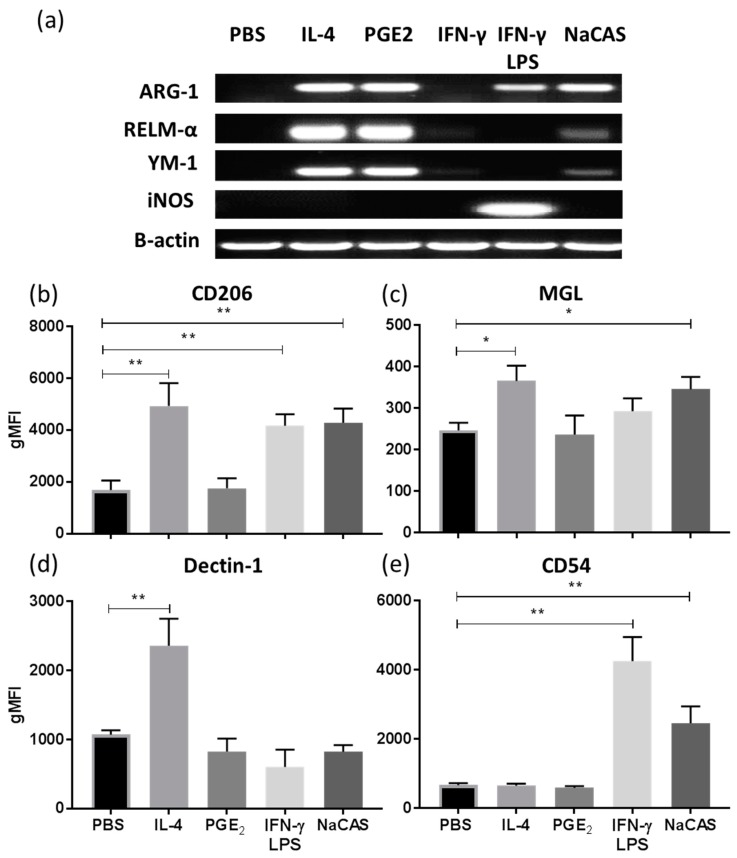
BMMφ were treated with NaCAS (1 mg/mL), M2a stimulant; IL-4 (20 ng/mL), M2c stimulant; PGE_2_ (5 μM), M1 stimulant; IFNγ (20 ng/mL) or IFNγ & LPS (100 ng/mL) for 18 h. PBS was used as an undifferentiated control. RNA was extracted to measure Arg-1, RELM α, Ym-1, iNOS and β-actin gene expression (**a**). The figure is representative image of three independent experiments. Treated BMMφ were also stained with specific antibodies for CD206 (**b**), MGL (**c**), Dectin-1 (**d**), CD54 (**e**) or with an isotype matched control and analysed by flow cytometry. Results were expressed as the geometrical mean ± SD of 3 independent experiments. *p*-values were calculated using one-way ANOVA. *, *p* ≤ 0.05, **, *p* ≤ 0.01 compared to the PBS control group.

**Figure 3 nutrients-11-01688-f003:**
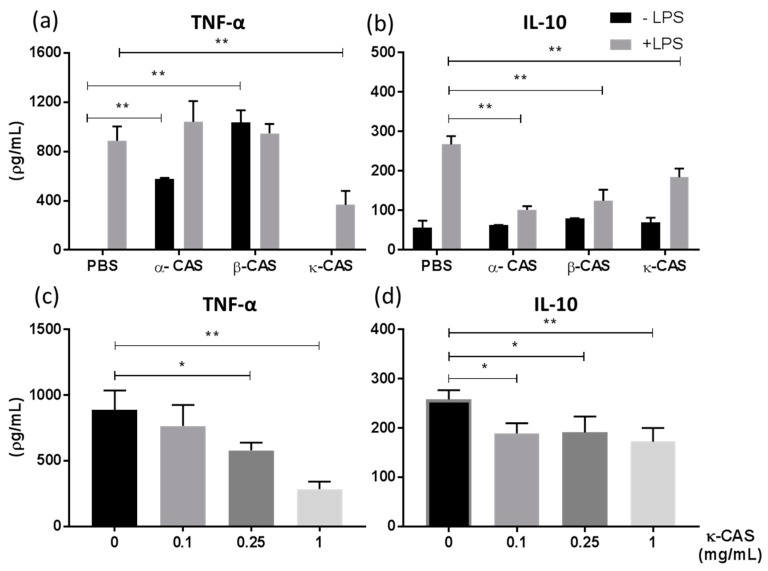
BMMφ were pre-treated with α-, β- or κ-CAS (1 mg/mL) and subsequently stimulated in the presence or absence of LPS (100 ng/mL) for 18 h and analysed for the secretion of TNF-α (**a**) or IL-10 (**b**) by ELISA. Results are expressed as mean ± SD of three independent experiments. *P*-values were calculated using two-way ANOVA. **, *p* ≤ 0.01 compared to PBS control group. BMMφ were also pre-treated with κ-CAS (0.1–1 mg/mL) and subsequently stimulated with LPS (100 ng/mL) for the secretion of TNF-α (**c**) and IL-10 (**d**) by ELISA after 18 h. Results are expressed as mean ± SD of three independent experiments. *p*-values were calculated using one-way ANOVA. *, *p* ≤ 0.05, **, *p* ≤ 0.01, compared to the PBS control group.

**Figure 4 nutrients-11-01688-f004:**
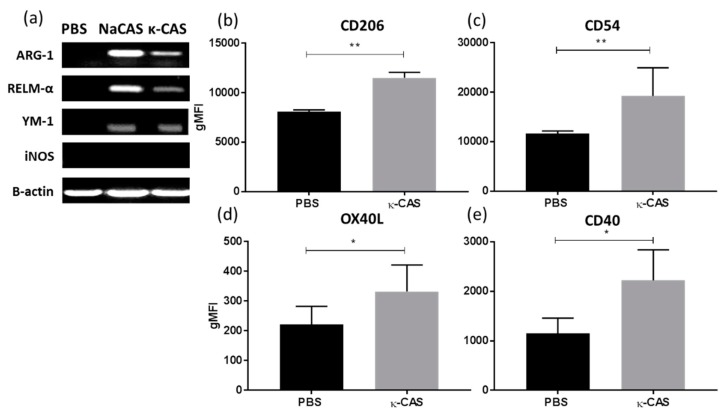
(**a**) RNA was extracted from PBS, NaCAS (1 mg/mL), or κ-CAS (1 mg/mL) treated cells after 18 h to measure the gene expression of Arg-1, RELM α, Ym-1, iNOS and β-actin by RT-PCR. The figure is representative of three independent experiments. κ-CAS (1 mg/mL) or PBS treated cells were also stained for 30 min with specific antibodies for CD206 (**b**), CD54 (**c**), OX40L (**d**), CD40 (**e**) or with an isotype matched control and analysed by flow cytometry. Results were analysed using FlowJo software and are expressed as the geometrical mean ± SD of at least three independent experiments. *p*-values were calculated using student’s *t* tests. **, *p* ≤ 0.05, **, *p* ≤ 0.01 compared to the PBS control group.

**Figure 5 nutrients-11-01688-f005:**
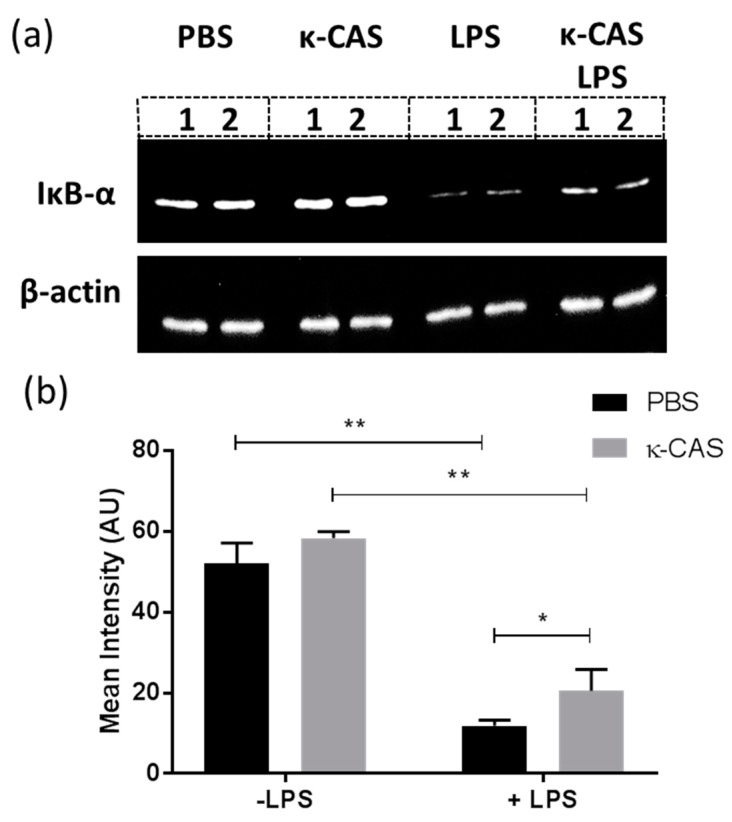
BMMφs were pre-treated with κ-CAS (1 mg/mL) for 2.5 h prior to stimulation with LPS (100 ng/mL) for 15 min. Control BMMφs were treated with PBS, κ-CAS or LPS alone. IκB-α protein levels was determined in whole-cell lysates by western blot analysis. A representative blot of each stimulation in duplicate is shown (**a**). Densitometric analysis was performed on all immunoblots, and IκB-α protein levels normalized using the housekeeping control protein β-actin and expressed as the average mean intensity in arbitrary units (**b**). Results are expressed as mean intensity ± SD of 3 independent experiments. *p*-values were calculated using two-way ANOVA. *, *p* ≤ 0.05, the LPS group compared to the pre-treated κ-CAS & LPS samples; **, *p* ≤ 0.01, the LPS treated group compared to non-LPS treated controls.

**Figure 6 nutrients-11-01688-f006:**
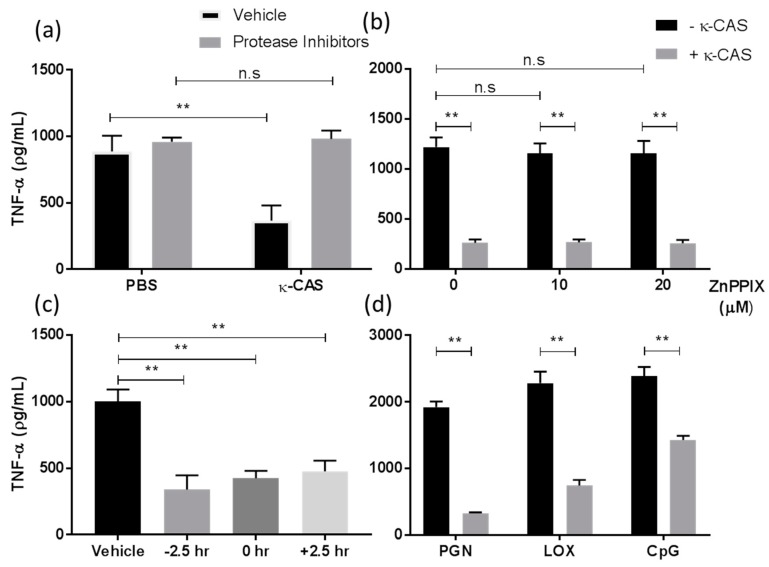
BMMφ were pre-treated with a protease inhibitor cocktail (1:500 *v*/*v*) and subsequently incubated with or without κ-CAS (1 mg/mL). Following incubation, cells were stimulated in the presence or absence of LPS (100 ng/mL) and supernatants were analysed for the secretion of TNF-α (**a**) by ELISA. *p*-values were calculated using two-way ANOVA. *, *p* ≤ 0.05, **, *p* ≤ 0.01 compared to PBS control group. BMMφ were also pre-treated with PBS or the heme oxygenase-1 inhibitor ZnPPIX (10 μM or 20 μM) prior to κ-CAS (1 mg/mL) treatment, followed by stimulation with LPS (100 ng/mL) and analysed for the secretion of TNF-α (**b**) by ELISA. Results are expressed as mean ± SD of three independent experiments. *p*-values were calculated using two way ANOVA. **, *p* ≤ 0.01, compared to κ-CAS treated control; n.s, compared to PBS treated controls. BMMφ treated with κ-CAS (1 mg/mL) 2.5 h prior, at the same time as, or 2.5 after hours post LPS (100 ng/mL) stimulation were analysed for the secretion of TNF-α (**c**) by ELISA. Results are expressed as mean ± SD of three independent experiments. *p*-values were calculated using one-way ANOVA multiple comparisons test. **, *p* ≤ 0.01 compared to the PBS control group. (D) BMMφ pre-treated κ-CAS (1 mg/mL) and subsequently stimulated in the presence or absence of PGN (5 μg/mL), LOX (0.5 mM) or CpG (2 μM) were analysed for the secretion of TNF-α (**d**) by ELISA. Results are expressed as mean ± SD of two independent experiments. *p*-values were calculated using multiple student’s t tests. *, *p* ≤ 0.05 **, *p* ≤ 0.01 compared to the PBS control group.

**Figure 7 nutrients-11-01688-f007:**
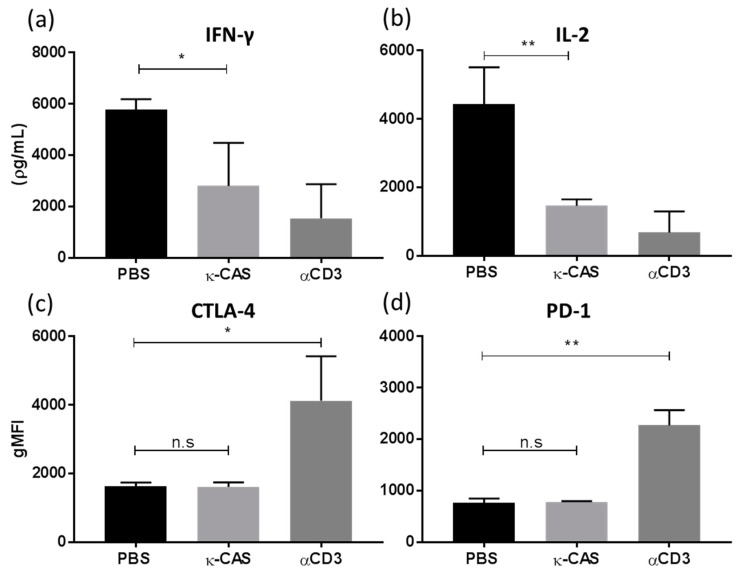
BMMφs pre-treated with κ-CAS (1 mg/mL) or PBS were co-cultured with CD4^+^ T-cells at a ratio of 1:4 on plates pre-coated with anti-CD3 (1 μg/well). CD4^+^ T-cells not cultured with BMMφ were used a negative control. Supernatants were analysed for the cytokines IFN-γ (**a**) and IL-2 (**b**) by ELISA. Results are expressed as mean ± SD of at least three independent experiments. *p*-values were calculated using one-way ANOVA *, *p* ≤ 0.05; **, *p* ≤ 0.01 compared to PBS control group. Following co-culture, CD4^+^ T-cells cells were analysed for the anergic extracellular surface markers CTLA4 (**c**) or PD-1 (**d**) by flow cytometry. Results were expressed as the geometrical mean ± SD of two independent experiments. *p*-values were calculated using one-way ANOVA. *, *p* ≤ 0.05, **, *p* ≤ 0.01 compared to PBS control group.

**Figure 8 nutrients-11-01688-f008:**
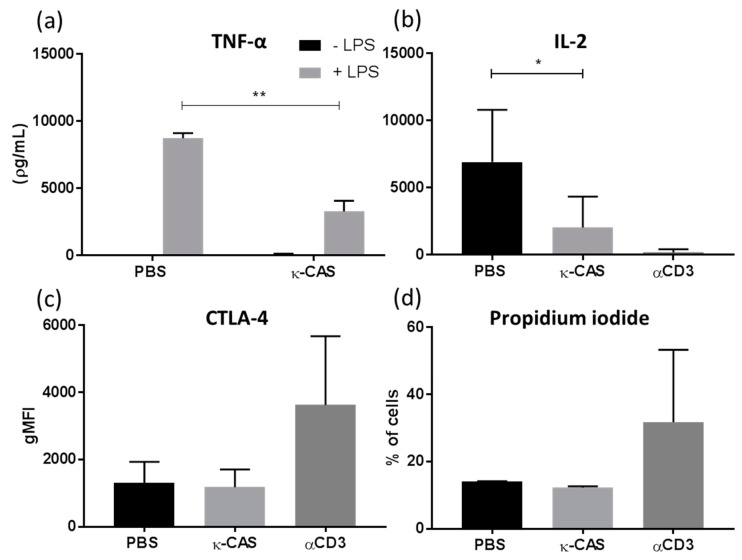
hMφ derived from CD14^+^ monocytes were pre-treated with κ-CAS (1 mg/mL) or PBS and subsequently stimulated in the presence or absence of LPS (100 ng/mL). Supernatants were analysed for the secretion of TNF-α (**a**) by ELISA. Results are expressed as mean ± SD from three individual donors. *p*-values were calculated using were calculated using two way ANOVA. **, *p* ≤ 0.01, compared to PBS control group. κ-CAS (1 mg/mL) or PBS treated monocytes were also co-cultured with CD4^+^ T-cells at a ratio of 1:10 on plates pre-coated with anti-CD3 (1 μg/well). CD4^+^ T-cells not cultured with cells were used a negative control. Supernatants were analysed after 72 h for the secretion of IL-2 (**b**) by ELISA. Results are expressed as mean ± SD from 4 donors. *p*-values were calculated using one-way ANOVA. *, *p* ≤ 0.05 compared to PBS control. Following co-culture, CD4^+^ T-cells cells were analysed by flow cytometry for the expression of the anergic extracellular surface marker; CTLA4 (**c**) and cell viability by measuring propidium iodide uptake (**d**). Results were expressed as the geometrical mean or percentage of cells positive ±SD from two donors.

**Table 1 nutrients-11-01688-t001:** Sense and anti-sense sequences of primers used for polymerase chain reaction (PCR) analysis.

Gene	Sense	Anti-Sense
***Arg-1***	CAGAAGAATGGAAGAGTCAG	CAGATATGCAGG GAGTCACC
***Ym-1***	TCACAGGTCTGGCAATTCTTCTG	TTTGTCCTTAGGAGGGCTTCCTC
***iNOS***	CCCTTCCGAAGTTTCTGGCAGCAGC	GGCTGTCAGAGAGCCTCGTGGCTTTGG
***RELMα***	GGTCCCAGTGCATATGGATGAGACCATAGA	CACCTCTTCACTCGAGGGACAGTTGGCAGC
***B-actin***	TGGAATCCTGTGGCATCCATGAAAC	TAAAACGCAGCTCAGTAACAGTCCG

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
