# Peer review of "Bovine κ-Casein Fragment Induces Hypo-Responsive M2-Like Macrophage Phenotype"

_nutrients, 2019, doi:10.3390/nu11071688_

Reviewer 1 Report

The work is very interesting and well prepared. 

Only one minor comment. Several parts are written two or three times in different parts for example: Lines 401-411. This part has been presented in the introduction.

Please correct all these parts in the whole manuscript.

Author Response

The initial paragraph to the discussion has been removed as suggested, as it repeats what was already highlighted in the introduction.  (Lines 401-411) 

Some other sentences have also been removed from the discussion that has been repeated elsewhere. (Lines 467-468; 440-441; 473-474)

Others however have been left in as we feel that the repeat of these sentences highlight essential differences in what is currently in the literature and the novelty of the findings in this study.  

Reviewer 2 Report

Interesting findings. This manuscript will be an important addition to current literature.

Author Response

No suggested changes 

Reviewer 3 Report

Excellent and thorough experimental setting. Promissing for future application on immunomodulatory neutraceuticals. 

Author Response

No suggested changes